# Self-Reported School Difficulties and the Use of the School Nurse Services by Adolescent Students

**DOI:** 10.3390/children8080647

**Published:** 2021-07-28

**Authors:** Siru Suoniemi, Anja Rantanen, Anna-Maija Koivisto, Katja Joronen

**Affiliations:** 1Unit of Health Sciences, Tampere University, Kalevantie 4, 33014 Tampere, Finland; siru.suoniemi@tuni.fi (S.S.); anja.rantanen@tuni.fi (A.R.); anna-maija.koivisto@tuni.fi (A.-M.K.); 2The School of Well-Being and Health Technology, Tampere University of Applied Sciences, Kuntokatu 3, 33520 Tampere, Finland; 3Department of Nursing Science, Turku University, 20014 Turku, Finland

**Keywords:** school difficulties, survey, adolescents, school health nurse services

## Abstract

Adolescents are increasingly finding school difficult and physical, mental and social problems increase the risk of exclusion. School health services help to identify problems and prevent them from escalating and the school nurse should be consulted when children are struggling academically. This study explored associations between school difficulties and the use of school health nurse services by 73,680 comprehensive school students with median age of 15.3. The study was based on nationally representative data from the 2017 Finnish School Health Promotion study and analyzed by gender. Difficulties in schooling were common and ranged from 9.9–32.7%. Girls reported difficulties more frequently than boys. Having self-reported difficulties was associated with greater use of school health nurse services, with girls seeking help more often than boys with similar issues and more boys saying they had no need for services. In addition, more self-reported difficulties with schooling were associated with unmet need for school health nurse services. School difficulties were associated with greater use of the school health nurse service use when the data were controlled for background factors. This study highlights shortcomings in access to school health nurse services by children with self-reported school difficulties and that girls were more likely report problems and seek help.

## 1. Introduction

The term learning difficulties is used to describe academic problems of different origins. It comprises general learning disabilities, low academic performance and special learning disabilities. Learning disabilities are disorders that affect an individual’s ability to learn and the problems they cause can include speaking, reading, writing, and paying attention [1]. Special learning disabilities refer to developmental or particular congenital difficulties [2]. The prevalence of Finnish children and adolescents with different learning difficulties has varied from 5–20% in different studies. Learning difficulties have increased and they have played a major role in social exclusion [3]. Identifying learning difficulties is important, because they can lead to poor learning outcomes and, at worst, dropping out of school [3,4]. The well-being of adolescents is a prerequisite for learning, which promotes the well-being of adolescents in many ways [5,6].

Learning disabilities and school difficulties have been found to be associated with a number of problems, including anxiety, depression, suicide attempts and violent behavior [7,8,9,10,11]. The social, physical and mental health and development of adolescents with learning disabilities and attention disorders should be individually assessed and supported. It is also important to pay attention to the protective factors for school difficulties [9]. Reading difficulties can have a negative impact on students and manifest as depression, anxiety, self-esteem problems and difficulty paying attention and concentrating [12]. Students with attention and behavioral disorders are more likely to have issues like mental health problems [13] and learning difficulties [13,14]. Emotional problems have also been associated with learning difficulties [14].

According to previous studies, the well-being, health and education of adolescents differ between genders [15,16,17]. Differences in the pressure students feel due to schoolwork increase with age, with 15-year-old girls reporting higher levels of pressure than boys [6]. Gender differences exist in learning, competence, motivation [18,19] and school success [18,20]. Boys have more reading difficulties and they are more likely to be permanent than in girls. Poor literacy, especially in boys, predicts poor school performance and has far-reaching negative effects [18]. Educational evaluation studies conducted in Finland showed Finnish girls performed better than boys in most areas that were evaluated. The three-yearly Programme for International Student Assessment (PISA) has come to the same conclusion about 15-year-old girls. Girls also have better grades at comprehensive school, which covers 7 to 16 year of age in Finland, than boys. As a result, more girls end up going to high school, boys end up pursuing vocational education. In terms of grades and assessment results, the differences have been particularly large in reading literacy [21,22]. European studies have found that girls are better readers than boys [23]. The 2015 PISA survey was the first time that Finnish girls were also more successful than boys in mathematics [21,24].

Gender differences have been shown to affect school performance and the grades that are achieved [20]. Gender also plays a role in school dropouts and the causes vary in different countries [25]. In Finland, most adolescents who leave school without a comprehensive school certificate at the age of 16 are boys. For example, boys accounted for 63% of the 510 school dropouts in 2017/2018. More Finnish boys than girls also drop out of secondary education, which usually lasts for two to four years [26]. According to data from the Organisation for Economic Co-operation and Development (OECD), 5.0% of males and 4.1% of females aged 15–19 were not in employment, education or training in Finland in 2019 [27]. One study found that boys were more likely than girls to report poor school satisfaction, school bullying, morning fatigue, dissatisfaction with school rules and the way that teachers treated them [28]. The girls in another study suffered from self-reported mental symptoms more frequently than boys [29]. Gender differences in the use of services have also been reported. Boys have reported difficulties accessing school nurses, even when resources were in line with recommendations [30]. Girls seemed to be more receptive to the help offered and were more likely to seek help [19]. These differences may be due to biological reasons, differences in access to services and the attitudes of healthcare professionals towards different genders [28].

School health nurses play an important and diverse role in the comprehensive school health services and student care that adolescents receive and they can have a direct impact on their health and well-being [31,32]. Their role includes assessing whether students need support and providing guidance for other experts [33]. The health of an adolescent can have a direct, or indirect, effect on whether they drop out of school or are excluded. School health professionals can improve students’ health and prevent the risk of exclusion [34]. In Finland, student welfare services promote student well-being on a community and individual basis. The services are universal and free of charge. The services they provide include school health care services and psychological and social work services [35]. These can just be provided by one professional or by multi-professional cooperation [36]. School health services are part of universal preventive health care. Services are organized municipally and include health examinations, health care counselling, acute medical care and promotion of well-being and safety at school [37,38].

Although school health nurse services are provided nationally in most countries, there are significant variation between countries when it comes to what they provide and how they support students. For example, an international review found that school health nurse services did not pay enough attention to mental health problems, even though these are one of the most common problems in adolescence [39]. Inequalities in the well-being, health and education of children and adolescents have been identified in Europe, based on socioeconomic backgrounds [6,40]. In Finland, students with an immigrant background, lower school well-being, lack of study support and greater health needs have reported more difficulties in access to school health nurses [30]. Adolescents with disabilities, such as difficulties with learning or concentrating, were more likely to be left without support and help from school and student health care services than other adolescents. In addition, some of the support and assistance needs that adolescents with disabilities have may not be identified [41]. Few studies have reported the associations between adolescents’ self-reported difficulties at school and their use of school health services. This study brings new knowledge on the subject and explores the understudied differences between boys and girls.

The aim of this study was to explore the association between self-reported school difficulties and the use of school health nurse services by 8th and 9th grade comprehensive school students (typically aged 14–16 years). We also assessed the confounding roles of background factors. 

## 2. Materials and Methods

### 2.1. Data and Participants

This was a quantitative cross-sectional study that was based on nationally representative data from the 2017 School Health Promotion study. Conducted by the Finnish Institute for Health and Welfare (THL) every second year, it focuses on Finnish children and adolescents and looks at their well-being, health, schoolwork and any assistance and services they receive to meet their needs [42]. The research material that we used was based on the responses from 8th and 9th grade students, who were attending comprehensive school. 

The data were collected in the schools between 1 March and 31 May 2017 [43]. The School Health Promotion study is completed anonymously, participation is voluntary, and the questionnaire is completed during a school class under the supervision of the teacher. It was possible to respond using a paper or digital version [44]. Respondents were given instructions and a username and password that could not be combined with personal information. One 45-min lesson was set aside to answer the questionnaire and additional time was provided if necessary [45]. The number of 8th and 9th grade students who participated in the 2017 survey was 73,680 (63%) [46].

### 2.2. Measures

The dependent variable in this study was the use of school health nurse services and the independent variable was difficulties in schooling. The background factors included demographic and socioeconomic characteristics: gender, grade level, parents’ education and employment, the family’s financial situation and respondent’s country of birth (Table 1).

The use of the school health nurse services was measured by asking whether they had visited them during the current school year, which started in autumn. We only wanted to know about visits that were in addition to their regular annual check-up, where they received health counseling, discussed matters that were on their mind received follow ups on previously identified issues [37]. The possible answers were that there was no need for additional visits, they tried to see the nurse but could not get an appointment, they had visited 1–2 times, 3–5 times or more than five times. If students said they were unable to get an appointment, then this this was categorized as unmet need for help and support [47]. 

Schooling difficulties were explored using an eight-item questionnaire. The students were asked whether they were experiencing difficulties in three school-related areas. 

The first was self-reported *learning difficulties* which comprised six items, namely difficulties in following the teaching in class, doing homework or other school tasks, preparing for exams, and performing tasks that required writing, reading and calculations. The second was *difficulties in oral presentations* and the third was *difficulties in using study devices,* namely digital technology and software. The students reported any individual difficulties using a four-point Likert scale that represented not at all, fairly little, quite a lot or very much.

### 2.3. Statistical Analysis

The chi-square test was used to analyze the associations between gender and background characteristics, prevalence of schooling difficulties and the use of school nurse health services. This revealed significant gender differences and we decided to analyze the data by gender. These were then described using frequencies and percentages. The association between schooling difficulties and the use of school nurse services was cross-tabulated and statistical significance was tested using the chi-square test.

A univariate multinomial regression analysis was used to explore the association between schooling difficulties and the use of school nurse services. We also carried out a multivariate multinomial regression analysis to identify the association between self-reported schooling difficulties and the use of nurse services when the data were controlled for background factors.

Background factors were selected based on previous studies and these included socio-economic background, which has been associated with adolescent’s health and education [6,40]. The other background factors included the parents’ education, whether they were unemployed during the last 12 months and respondent’s country of birth, which was either Finnish or foreign-born. The multinomial regression analysis looked at four categories of school health nurse visits. The categories 3–5 times and more than five times were combined to form the category more than two times. The other three categories were kept as they were: there was no need for to visit the nurse, they student could not get an appointment and the student saw the nurse 1–2 times. In addition, we combined the self-reported difficulties in schooling categories of quite a lot and very much. The other two categories, not at all and fairly little, were kept as they were.

Following the multinomial regression analysis, the level of association between schooling difficulties and the use of school nursing services was described by odds ratios (ORs) and their 95% confidence intervals (CI) [48]. The reference categories were no need to use the school health nurse service and not at all for schooling difficulties. The level of statistical significance was set at *p* < 0.01 due to the large dataset [49]. SPSS Statistics, version 25 (IBM Corp, New York, NY, USA) was used to analyze the data.

### 2.4. Ethical Considerations

Responsible research conduct was followed during the study [50]. This refers to integrity, meticulousness and accuracy when conducting research and reporting, presenting and evaluating the results. The 2017 data collection was approved by the ethical committee of the Finnish Institute for Health and Welfare (number THL/1704/6.02.0 1/2016) [44]. The students and their parents or guardians were informed about the study by a letter that emphasized the anonymous and voluntary nature of the study. 

## 3. Results

### 3.1. Participants in the Study

A total of 73,680 adolescents participated in this study and Table 1 shows their sociodemographic characteristics. The gender distribution was even, with 50.1% girls and 49.2% boys. Some respondents (0.7%) did not report their gender and these 521 respondents were excluded from the gender analysis. There were almost as many students in the 8th (49.4%) and 9th (49.9%) grades, with 0.4% not answering this question. Their median age was 15.3 (Q1 14.8 and Q3 15.8).

### 3.2. Self-Reported Difficulties in Schooling

Difficulties in schooling were examined by eight self-reported areas. When it came to learning difficulties, boys had slightly more difficulties in doing homework and other school tasks, with 21.9% of boys answering that it was quite or very difficult, compared to 20.9% of the girls. The respective responses for writing tasks were 20.9% for the boys and 19.1% for the girls (*p* < 0.001). With regard to other self-reported learning difficulties, girls (19.0–33.8%) had more difficulties than boys (16.0–29.6%) (*p* < 0.001) (Table 2).

The differences between the difficulties in oral presentations and using study devices experienced by the girls and boys were statistically significant (*p* < 0.001). For example 39.0% of girls and 26.5% of boys answered quite a lot or very much when asked about whether they experienced difficulties in oral presentations (*p* < 0.001) and the figures were 10.8% and 8.9%, respectively, for difficulties in using study devices (*p* < 0.001) (Table 2).

### 3.3. Use of School Health Nurse Services

More than a third (36.0%) of the respondents had visited a school health nurse for some other reason than their regular annual checkup and most (26.7%) had used the services 1–2 times. There was a statistically significant gender difference in the frequency of the visits (*p* < 0.001). Girls used the school health nurse service more often than boys and it was twice as much in those who had visited the nurse 3–5 times or more than five times. In addition, 2.2% had attempted to get an appointment, but couldn’t and this rate was slightly higher in girls than boys (Table 3).

### 3.4. The Association between Self-Reported School Difficulties and the Use of School Health Nurse Services

All school difficulties related to the use of school health nurse services were statistically significant (*p* < 0.001) in both girls and boys. Students with more self-reported difficulties used more school health nurse services than respondents with none of fairly little difficulties. Table 4 and Table 5 present the separate data for girls and boys.

Multinomial regression analysis was used to examine the association between school difficulties and the use of school health nurse services by using a univariate model, without background factors. A multivariate model was constructed to control the background factors of the respondents, their parents’ education, any unemployment during the last 12 months and their country of birth.

These multinomial regression analysis models were conducted separately by gender and are presented using univariate and multivariate odds ratios (OR) and 95% confidence intervals (CI) for girls (Table 6) and boys (Table 7). They examine school health nurse services, difficulties in schooling and possible confounding factors. The data in these two tables show statistically significant associations between all school difficulties and the use of school health nurse services. The multivariate ORs for those who used services more than twice show that more difficulties were associated with a greater use of services. Girls who found it quite or very difficult to following class lessons had visited a school health nurse more often than those girls who had no difficulties (OR 3.79, 95% CI 3.44–4.18). The respective figures were also higher for the boys who experienced difficulties during classes (OR 2.84, 95% CI 2.45–3.28).

Respondents who had been unable to get an appointment with the school nurse service reported more difficulties than those who had seen the nurse. Girls who found it quite or very difficult to follow class lessons were more likely to say they had not been able to see the school health nurse than those who had no difficulties in that area (OR 2.71, 95% CI 2.21–3.32). The same was true for the boys (OR 2.33, 95% CI 1.84–2.95).

## 4. Discussion

This study explored the association between self-reported school difficulties and the use of school health nurse services among Finnish comprehensive school students. The data source was the 2017 THL School Health Promotion study.

We found that 9.9–32.7% of the students had experienced different school difficulties quite a lot or very much. The use of digital study devices and software caused the least difficulties for the respondents, as only 9.9% of the students found it quite or very difficult to use these devices. Furthermore, 17.6–32.7% of all respondents said that they found other areas of schooling quite or very difficult. A previous study indicated that 5–20% of Finns had learning difficulties, but the prevalence has varied between studies [3]. This may have been due to the definition, delineation, and measurement of the concept of learning disabilities. In the present study, the students self-reported any difficulties in schooling and learning, but their assessment of what those terms meant did not necessarily meet the diagnostic criteria of the classifications in the International Classification of Diseases, Tenth Revision. Our results showed a higher number of self-reported difficulties in most of the areas of schooling than in previous studies that identified rates of learning difficulties. A lot of the difficulties that the students reported to the 2017 survey may affect their schooling in the future. School difficulties may also be linked to students’ self-esteem and support would also help them to trust their own skills and abilities.

This study indicated that self-reported difficulties were different between girls and boys. Boys had slightly more difficulties with homework or other school tasks, and writing tasks, than girls. Girls had more problems with the other schooling difficulties that were studied. For example, girls had significantly more difficulties with oral presentations than boys. Gender differences related to school difficulties and education have been reported by other studies. 

Previous studies showed that boys had more reading difficulties than girls [18]. Finnish educational evaluation studies and PISA evaluations have also shown that Finnish girls were more successful than boys in most areas [21]. Schooling difficulties were self-reported in this study, while other studies, including the PISA surveys, measured competencies using various indicators and tests, so do our results show that girls have more difficulties with schooling or are they just more likely to actually report their difficulties? Is it possible that girls are more motivated at school [19] and have better competency results [21,22,24], which makes schooling more goal-oriented and burdensome? And do the higher expectations that they have for themselves make schooling more challenging and difficult?

According to our study, 26.0% of the girls who found it very much difficult to perform tasks that required reading had visited the school health nurse more than twice. Those girls who had no problems at all in this area 10.2% had visited a nurse more than twice. This number did not include their regular annual check-up. The respective percentages for the boys were 13.5% and 4.7%. We also found that 4.1% of the boys who found tasks involving reading very difficult had not been able to get an appointment to see the nurse. Literacy plays a very important role in school success. A previous study found that poor literacy, especially in boys, predicted far-reaching negative effects in the future [18].

According to our results, boys used, or received, school health nurse services less than girls, apart from the regular annual check-ups. Girls visited the school nurse more than boys with similar difficulties. This study raises questions about why there was a gender gap in service use. The boys suggested that they did not need help. Boys who found schooling quite or very difficult were more likely to say they had no need to see the school health nurse than girls in the same position. Were the girls’ difficulties better identified than the boys’ difficulties and did that make it easier to help them? Or are girls more active when it comes to seeking help? And do boys receive support and assistance from other professionals than from the school health nurse. These differences may also be due to what help they receive. However, we do not know for what purpose do the respondents visited the school health nurse. Previous research suggested that girls were more receptive to the help they were offered and were more likely to seek help [19]. 

There was a statistically significant association between self-reported schooling difficulties and the use of the school health nurse services. Students who had more difficulties tended to use more services than those who had fewer difficulties. On the one hand, this is good, because student welfare services can help adolescents when they have difficulties [51]. According to previous research, learning and concentration difficulties at school have a negative impact on adolescents’ physical, mental and social development [7,9,12,13]. Adolescents with learning difficulties and attention disorders need multi-professional and individual support [13,52]. On the other hand, if student welfare services have to deal with lots of schooling difficulties, this increases concerns about resources, the adequacy of services and equal access. Previous reports, such as the 2015 report from the Finnish Regional State Administrative Agency 2015 [53], and Finnish studies have also raised concerns about the adequacy and equality of student care services [30]. The Finnish National Program for Youth Work and Policy for 2020–2023 has proposed adequate resources for study provision and coordinated co-operation between professional groups [5].

Our results highlighted shortcomings in access to services and equal distribution. More self-reported schooling difficulties were associated with a greater use of school health nurse services and lower access to services. Girls used the services more than boys. Previous research has shown clear differences in the organization and availability of student care services among regions and schools of different sizes [54,55]. This means that students do not enjoy the same access. Student welfare services can decisively support and promote the balanced growth and development of students and prevent problems from starting and developing. This can help to reduce exclusion [51]. School health nurses should get involved when students have schooling difficulties and general support is not enough to help them with learning [3].

According to our results, students who had a number of schooling difficulties were more likely to be left without help and support from school health nurse than students who only had a few difficulties. Self-reported schooling difficulties were associated with lower service needs being met. The results were consistent with a previous study [41] that showed that adolescents with disabilities had a higher risk of being left without support and help from school and student health care than other adolescents. There are clearly wide differences in access to support between people with disabilities and other adolescents and these lead to inequalities [41].

## 5. Strengths and Limitations

We used nationally representative data from the 2017 Finnish School Health Promotion study, which increased the external validity of the study. The questionnaire was anonymous and voluntary, it was completed during the school day and 63% of all Finnish 8th and 9th grade students took part. Most municipalities in Finland participated in the survey, so the material is comprehensive throughout the country. In Finland school health care and comprehensive school are free of charge, so there are no major differences in participation in education or the provision of services in different socio-economic groups. Additionally, the survey was available in different languages, in Finnish, Swedish, English, Russian and North Sami.

Some limitations should be considered. Poor school satisfaction or motivation may have influenced how willing some students were to respond. Also, the reliability of research may have been undermined by the self-reports, which were the students’ subjective experiences of school difficulties. Additionally, students who were not present at school on the data-gathering day were not able to participate in the study. Furthermore, students with major difficulties in reading or understanding the common language responded to a shorter questionnaire in plain language; this group of students were excluded from this study. A further limitation was the substantial amount of missing values in some variables; e.g., 12.7 % of students did not report father’s education. 

## 6. Conclusions

This nationally representative study indicated that the proportion of self-reported school-related difficulties among 8th and 9th grade students was high in Finland. We also showed that more self-reported schooling difficulties were associated with a greater use of school health nurse services and lower access to services in some cases. This highlighted issues with the availability, and equal distribution, of school health nurse services across the country. The findings highlight an urgent need for students to have equal access to school nurse services and that the services need sufficient resources to support students with schooling difficulties. Future research is needed to explore the experiences of school nurses with regard to the services they provide for girls and boys. In addition, further research should be carried out with those students whose service needs are not being met.

## Figures and Tables

**Table 1 children-08-00647-t001:** Background characteristics of students who took part in the THL School Health Promotion study 2017.

Background	All (*n* = 73,680) % (*n*)	Girls (*n* = 36,883) % (*n*)	Boys (*n* = 36,276) % (*n*)
Gender			
Boy	49.2 (36,276)		
Girl	50.1 (36,883)		
Missing	0.7 (521)		
Grade level			
8th grade	49.9 (36,788)	50.0 (18,426)	50.0 (18,145)
9th grade	49.7 (36,625)	49.9 (18,389)	49.8 (18,050)
Missing	0.4 (267)	0.2 (68)	0.2 (81)
Mother’s education			
Comprehensive school or equivalent	5.7 (4226)	6.0 (2202)	5.5 (1996)
Upper secondary school, high school or vocational education	25.8 (19,033)	26.2 (9652)	25.6 (9274)
Occupational studies in addition to upper secondary school, high school, or vocational education	21.4 (15,737)	22.8 (8426)	19.9 (7236)
University, university of applied sciences or other higher education	36.1 (26,569)	37.3 (13,740)	35.0 (12,693)
Missing	11.0 (8115)	7.8 (2863)	14.0 (5077)
Father’s education			
Comprehensive school or equivalent	8.1 (5940)	8.4 (3084)	7.8 (2816)
Upper secondary school, high school or vocational education	29.5 (21,766)	30.4 (11,210)	28.8 (10,448)
Occupational studies in addition to upper secondary school, high school, or vocational education	19.3 (14,229)	20.6 (7582)	18.1 (6570)
University, university of applied sciences or other higher education	30.4 (22,371)	30.6 (11,301)	30.2 (10,955)
Missing	12.7 (9374)	10.0 (3706)	15.1 (5487)
Whether parents have been unemployed or laid-off during the past 12 months			
No	64.7 (47,662)	65.1 (24,022)	64.5 (23,392)
Yes, one parent	25.7 (18,908)	28.0 (10,323)	23.4 (8481)
Yes, two or more of the parents	3.7 (2708)	3.7 (1348)	3.7 (1332)
Missing	6.0 (4402)	3.2 (1190)	8.5 (3071)
Family’s perceived financial situation			
Very good	21.5 (15,872)	18.0 (6656)	25.2 (9124)
Fairly good	42.4 (31,253)	44.0 (16,220)	41.0 (14,879)
Moderate	23.7 (17,498)	27.3 (10071)	20.2 (7322)
Fairly poor	5.1 (3748)	6.3 (2338)	3.8 (1389)
Very poor	1.3 (946)	1.1 (424)	1.4 (507)
Missing	5.9 (4363)	3.2 (1174)	8.4 (3055)
Background by country of birth			
Finnish background	88.6 (65,317)	92.1 (33,981)	85.4 (30,995)
Foreign background	5.5 (4066)	5.0 (1832)	6.0 (2184)
Missing	5.8 (4297)	2.9 (1070)	8.5 (3097)
Lived in Finland			
All my life	86.9 (64,046)	90.0 (33,199)	84.1 (30,504)
More than 10 years, but not always	3.9 (2885)	4.0 (1470)	3.9 (1397)
5–10 years	2.0 (1450)	2.0 (750)	1.9 (683)
1–4 years	0.8 (589)	0.8 (302)	0.8 (280)
Less than 1 year	0.9 (688)	0.4 (138)	1.5 (539)
Missing	5.5 (4022)	2.8 (1024)	7.9 (2873)

*p*-value: grade level 0.827, other < 0.001.

**Table 2 children-08-00647-t002:** Self-reported difficulties in schooling among students who took part in the 2017 THL School Health Promotion study.

Variable	All (*n* = 73,680) % (*n*)	Girls (*n* = 36,883) % (*n*)	Boys (*n* = 36,276) % (*n*)
(1) Self-reported learning difficulties			
Difficulties in following the teaching in class			
Not at all	33.4 (24,573)	32.0 (11,808)	34.9 (12,654)
Fairly little	48.1 (35,412)	48.4 (17,847)	47.9 (17,373)
Quite a lot	14.2 (10,429)	15.4 (5674)	12.9 (4681)
Very much	3.4 (2503)	3.6 (1335)	3.1 (1126)
Missing	1.0 (763)	0.6 (219)	1.2 (442)
Difficulties in doing homework or other school tasks			
Not at all	34.8 (25,666)	36.5 (13,479)	33.3 (12,083)
Fairly little	42.6 (31,366)	41.9 (15,449)	43.4 (15,746)
Quite a lot	15.7 (11,569)	15.4 (5680)	16.0 (5803)
Very much	5.8 (4247)	5.5 (2038)	5.9 (2151)
Missing	1.1 (832)	0.6 (237)	1.4 (493)
Difficulties in preparing for exams			
Not at all	23.6 (17,387)	22.0 (8110)	25.3 (9193)
Fairly little	43.5 (32,071)	43.5 (16,051)	43.7 (15,863)
Quite a lot	23.5 (17,348)	24.9 (9177)	22.2 (8064)
Very much	8.2 (6028)	8.9 (3289)	7.4 (2671)
Missing	1.1 (846)	0.7 (256)	1.3 (485)
Difficulties in performing tasks that require writing			
Not at all	35.5 (26,127)	38.3 (14,124)	32.7 (11,864)
Fairly little	43.3 (31,897)	41.9 (15,441)	44.9 (16,302)
Quite a lot	15.2 (11,233)	14.6 (5396)	15.9 (5767)
Very much	4.8 (3509)	4.5 (1648)	5.0 (1807)
Missing	1.2 (914)	0.7 (274)	1.5 (536)
Difficulties in performing tasks that require reading			
Not at all	39.3 (28,943)	41.3 (15,225)	37.4 (13,585)
Fairly little	39.2 (28,916)	37.0 (13,630)	41.7 (15,134)
Quite a lot	14.9 (11,012)	15.3 (5659)	14.5 (5266)
Very much	5.3 (3888)	5.6 (2081)	4.9 (1762)
Missing	1.3 (921)	0.8 (288)	1.5 (529)
Difficulties in performing tasks that require calculation			
Not at all	34.9 (25,693)	31.9 (11,777)	38.0 (13,798)
Fairly little	40.8 (30,067)	40.3 (14,858)	41.5 (15,052)
Quite a lot	16.8 (12,409)	19.3 (7104)	14.4 (5229)
Very much	6.1 (4495)	7.6 (2806)	4.5 (1627)
Missing	1.4 (1016)	0.9 (338)	1.6 (570)
(2) Difficulties in oral presentations			
Not at all	25.1 (18,482)	21.8 (8024)	28.5 (10,342)
Fairly little	40.9 (30,155)	38.5 (14,210)	43.6 (15,804)
Quite a lot	20.8 (15,327)	23.3 (8582)	18.4 (6666)
Very much	11.9 (8784)	15.7 (5773)	8.1 (2927)
Missing	1.3 (932)	0.8 (294)	1.5 (534)
(3) Difficulties in using study devices			
Not at all	52.0 (38,317)	47.1 (17,378)	57.2 (20,741)
Fairly little	36.6 (26,988)	41.2 (15,188)	32.1 (11,655)
Quite a lot	7.2 (5288)	8.1 (2983)	6.2 (2265)
Very much	2.7 (1991)	2.7 (979)	2.7 (981)
Missing	1.5 (1096)	1.0 (355)	1.7 (634)

*p*-value: all variables < 0.001.

**Table 3 children-08-00647-t003:** Use of school health nurse services by students who took part in the 2017 THL School Health Promotion study.

Variable	All (*n* = 73,680) % (*n*)	Girls (*n* = 35,830) % (*n*)	Boys (*n* = 33,598) % (*n*)
Visited the school health nurse, other than for a regular check-up			
No, there was no need for it	56.5 (41,662)	50.8 (18,741)	62.6 (22,705)
No, I tried but could not get an appointment	2.2 (1632)	2.5 (911)	1.9 (704)
Yes, 1–2 times	26.7 (19,686)	30.7 (11,327)	22.8 (8256)
Yes, 3–5 times	5.5 (4018)	7.7 (2841)	3.2 (1154)
Yes, more than 5 times	3.8 (2819)	5.4 (2010)	2.1 (779)
Missing	5.2 (3863)	2.9 (1053)	7.4 (2678)

*p*-value < 0.001.

**Table 4 children-08-00647-t004:** Self-reported school difficulties and use of the school health nurse services by girls who took part in the 2017 THL School Health Promotion study.

	Visited the School Health Nurse, Other Than for a Regular Checkup				
	No, There Was No Need for It % (*n*)	No, I Tried but Could Not Get an Appointment % (*n*)	Yes, 1–2 Times % (*n*)	Yes, 3–5 Times % (*n*)	Yes, More Than 5 Times % (*n*)
(1) Self-reported learning difficulties					
Difficulties in following the teaching in class					
Not at all	60.3 (6977)	1.8 (211)	29.4 (3404)	5.6 (653)	2.9 (333)
Fairly little	51.1 (8898)	2.6 (456)	33.0 (5749)	8.1 (1403)	5.2 (902)
Quite a lot	42.5 (2326)	3.2 (177)	32.1 (1756)	11.7 (640)	10.4 (569)
Very much	37.6 (460)	5.2 (64)	30.2 (369)	11.0 (135)	15.9 (195)
Difficulties in doing homework or other school tasks					
Not at all	59.1 (7829)	1.9 (257)	29.7 (3936)	6.1 (814)	3.1 (412)
Fairly little	51.1 (7678)	2.7 (408)	33.2 (4999)	7.8 (1174)	5.2 (781)
Quite a lot	44.0 (2406)	2.9 (159)	32.7 (1787)	11.2 (610)	9.3 (508)
Very much	38.8 (737)	4.4 (83)	29.1 (553)	12.2 (232)	15.6 (296)
Difficulties in preparing for exams					
Not at all	60.6 (4822)	1.7 (139)	29.1 (2316)	5.8 (459)	2.7 (216)
Fairly little	53.9 (8468)	2.5 (392)	32.4 (5081)	7.1 (1110)	4.1 (646)
Quite a lot	46.1 (4093)	2.8 (246)	33.2 (2952)	10.1 (894)	7.9 (701)
Very much	40.5 (1259)	4.1 (128)	29.6 (922)	11.9 (370)	13.9 (432)
Difficulties in performing tasks that require writing					
Not at all	56.4 (7809)	2.2 (303)	30.6 (4234)	7.0 (968)	3.9 (543)
Fairly little	51.6 (7760)	2.5 (371)	32.9 (4943)	7.7 (1158)	5.4 (810)
Quite a lot	46.5 (2416)	3.2 (166)	31.9 (1656)	10.1 (522)	8.3 (431)
Very much	41.9 (648)	4.1 (64)	28.9 (447)	11.6 (180)	13.4 (208)
Difficulties in performing tasks that require reading					
Not at all	57.2 (8557)	2.1 (319)	30.5 (4565)	6.6 (988)	3.6 (535)
Fairly little	51.4 (6822)	2.6 (349)	32.7 (4335)	7.9 (1050)	5.4 (713)
Quite a lot	44.8 (2435)	3.1 (169)	33.2 (1804)	10.3 (560)	8.6 (469)
Very much	41.5 (807)	3.5 (68)	29.0 (565)	11.7 (227)	14.3 (278)
Difficulties in performing tasks that require calculation					
Not at all	57.6 (6644)	2.3 (266)	30.1 (3480)	6.5 (746)	3.5 (408)
Fairly little	52.1 (7556)	2.4 (343)	32.5 (4715)	7.7 (1119)	5.3 (766)
Quite a lot	47.8 (3278)	2.7 (186)	32.7 (2245)	9.5 (653)	7.3 (501)
Very much	42.0 (1121)	4.1 (109)	30.5 (814)	11.5 (306)	11.9 (316)
(2) Difficulties in oral presentations					
Not at all	51.8 (4047)	2.5 (195)	32.1 (2514)	8.1 (637)	5.5 (427)
Fairly little	53.7 (7446)	2.3 (323)	32.2 (4475)	7.2 (1006)	4.5 (627)
Quite a lot	53.4 (4448)	2.3 (194)	30.9 (2573)	7.8 (650)	5.6 (464)
Very much	48.1 (2690)	3.4 (191)	30.5 (1705)	9.6 (539)	8.4 (473)
(3) Difficulties in using study devices					
Not at all	53.5 (9094)	2.2 (380)	31.0 (5278)	7.9 (1342)	5.3 (907)
Fairly little	51.9 (7678)	2.5 (373)	32.5 (4809)	7.9 (1174)	5.2 (769)
Quite a lot	50.0 (1423)	3.4 (98)	31.3 (892)	8.0 (229)	7.2 (204)
Very much	44.3 (407)	5.3 (49)	28.8 (265)	8.9 (82)	12.6 (116)

*p*-value: all variables < 0.001.

**Table 5 children-08-00647-t005:** Self-reported school difficulties and use of the school health nurse services by boys who took part in the 2017 THL School Health Promotion study.

	Visited the School Health Nurse, Other Than for a Regular Checkup				
	No, There Was No Need for It % (*n*)	No, I Tried but Could Not Get an Appointment % (*n*)	Yes, 1–2 Times % (*n*)	Yes, 3–5 Times % (*n*)	Yes, More Than 5 Times % (*n*)
(1) Self-reported learning difficulties					
Difficulties in following the teaching in class					
Not at all	72.8 (8690)	1.5 (183)	21.7 (2585)	2.4 (284)	1.6 (195)
Fairly little	66.3 (10,804)	2.1 (336)	26.2 (4262)	3.6 (591)	1.8 (296)
Quite a lot	59.5 (2501)	2.9 (122)	28.1 (1182)	5.7 (238)	3.8 (159)
Very much	59.5 (573)	5.5 (53)	19.2 (185)	3.1 (30)	12.7 (122)
Difficulties in doing homework or other school tasks					
Not at all	72.7 (8288)	1.7 (195)	21.8 (2487)	2.3 (263)	1.4 (164)
Fairly little	66.9 (9867)	1.9 (279)	26.0 (3829)	3.5 (517)	1.7 (256)
Quite a lot	61.6 (3259)	2.7 (145)	27.2 (1439)	4.9 (261)	3.5 (186)
Very much	59.6 (1148)	3.9 (76)	22.8 (440)	5.1 (99)	8.5 (163)
Difficulties in preparing for exams					
Not at all	73.4 (6330)	1.6 (142)	20.9 (1806)	2.3 (200)	1.7 (148)
Fairly little	68.2 (10,134)	1.9 (284)	25.2 (3744)	3.2 (474)	1.6 (232)
Quite a lot	62.4 (4650)	2.4 (182)	27.6 (2057)	4.7 (348)	2.9 (214)
Very much	59.6 (1434)	3.6 (87)	24.6 (592)	5.0 (120)	7.3 (175)
Difficulties in performing tasks that require writing					
Not at all	70.7 (7908)	1.8 (196)	23.0 (2575)	2.7 (306)	1.7 (194)
Fairly little	67.7 (10,338)	1.9 (287)	25.3 (3877)	3.5 (535)	1.6 (246)
Quite a lot	63.2 (3337)	2.7 (143)	26.6 (1404)	4.3 (225)	3.3 (172)
Very much	59.5 (955)	4.1 (65)	22.2 (356)	4.6 (73)	9.7 (155)
Difficulties in performing tasks that require reading					
Not at all	70.6 (9072)	1.9 (243)	22.9 (2937)	2.9 (367)	1.8 (225)
Fairly little	67.9 (9592)	1.8 (251)	25.2 (3568)	3.3 (473)	1.8 (250)
Quite a lot	61.9 (2966)	2.7 (131)	27.3 (1309)	4.7 (227)	3.2 (155)
Very much	58.4 (915)	4.1 (64)	24.0 (376)	4.9 (76)	8.6 (135)
Difficulties in performing tasks that require calculation					
Not at all	70.8 (9248)	1.7 (221)	23.1 (3014)	2.8 (365)	1.7 (216)
Fairly little	66.9 (9403)	1.9 (266)	25.6 (3598)	3.7 (513)	1.9 (266)
Quite a lot	62.8 (2973)	2.9 (138)	26.9 (1273)	4.4 (207)	3.1 (145)
Very much	60.9 (884)	4.5 (65)	20.9 (303)	4.0 (58)	9.7 (141)
(2) Difficulties in oral presentations					
Not at all	67.9 (6539)	1.8 (176)	24.5 (2360)	3.4 (330)	2.4 (230)
Fairly little	68.6 (10,156)	2.0 (298)	24.5 (3627)	3.2 (480)	1.7 (249)
Quite a lot	66.7 (4138)	2.1 (129)	25.6 (1590)	3.5 (215)	2.1 (128)
Very much	63.5 (1708)	3.4 (91)	22.8 (614)	4.2 (114)	6.0 (162)
(3) Difficulties in using study devices					
Not at all	68.0 (13,389)	1.7 (334)	25.0 (4932)	3.4 (663)	2.0 (386)
Fairly little	67.8 (7288)	2.2 (241)	24.6 (2644)	3.4 (365)	2.0 (211)
Quite a lot	65.4 (1277)	3.6 (71)	23.6 (460)	4.0 (79)	3.4 (66)
Very much	61.1 (514)	5.5 (46)	17.7 (149)	3.6 (30)	12.1 (102)

*p*-value: all variables < 0.001.

**Table 6 children-08-00647-t006:** Univariate and multivariate odds ratios (OR) for girls, based on successful access to school nurse health services, schooling difficulties and possible confounding factors, as recorded in the 2017 School Health Promotion study 2017.

Visited the School Health Nurse, Other Than for a Regular Check-Up						
	Univariate Model			Multivariate Model		
	No, I Tried but Could Not Get an Appointment OR (95% CI)	Yes, 1–2 Times OR (95% CI)	Yes, More Than 2 Times OR (95% CI)	No, I Tried but Could Not Get an Appointment OR (95% CI)	Yes, 1–2 Times OR (95% CI)	Yes, More Than 2 Times OR (95% CI)
(1) Self-reported learning difficulties						
Difficulties in following the teaching in class						
Not at all (ref.)	1.00	1.00	1.00	1.00	1.00	1.00
Fairly little	**1.70 (1.44–2.00)**	**1.32 (1.26–1.40)**	**1.83 (1.69–1.99)**	**1.69 (1.42–2.01)**	**1.31 (1.24–1.38)**	**1.81 (1.66–1.97)**
Quite a lot or very much	**2.86 (2.37–3.46)**	**1.56 (1.46–1.68)**	**3.91 (3.57–4.28)**	**2.71 (2.21–3.32)**	**1.55 (1.44–1.68)**	**3.79 (3.44–4.18)**
Difficulties in doing homework or other school tasks						
Not at all (ref.)	1.00	1.00	1.00	1.00	1.00	1.00
Fairly little	**1.62 (1.38–1.90)**	**1.30 (1.23–1.37)**	**1.63 (1.50–1.76)**	**1.62 (1.38–1.92)**	**1.28 (1.21–1.36)**	**1.59 (1.47–1.73)**
Quite a lot or very much	**2.35 (1.96–2.81)**	**1.48 (1.39–1.58)**	**3.34 (3.07–3.64)**	**2.25 (1.85–2.73)**	**1.49 (1.38–1.59)**	**3.23 (2.95–3.54)**
Difficulties in preparing for exams						
Not at all (ref.)	1.00	1.00	1.00	1.00	1.00	1.00
Fairly little	**1.61 (1.32–1.96)**	**1.25 (1.18–1.33)**	**1.48 (1.35–1.63)**	**1.69 (1.38–2.08)**	**1.24 (1.17–1.32)**	**1.48 (1.34–1.64)**
Quite a lot or very much	**2.42 (1.99–2.96)**	**1.51 (1.41–1.61)**	**3.20 (2.91–3.51)**	**2.45 (1.98–3.03)**	**1.50 (1.40–1.61)**	**3.09 (2.79–3.42)**
Difficulties in performing tasks that require writing						
Not at all (ref.)	1.00	1.00	1.00	1.00	1.00	1.00
Fairly little	**1.23 (1.06–1.44)**	**1.18 (1.12–1.24)**	**1.31 (1.22–1.41)**	**1.27 (1.08–1.50)**	**1.16 (1.10–1.23)**	**1.28 (1.18–1.38)**
Quite a lot or very much	**1.94 (1.62–2.31)**	**1.27 (1.18–1.35)**	**2.26 (2.08–2.46)**	**1.82 (1.50–2.20)**	**1.24 (1.15–1.33)**	**2.13 (1.94–2.33)**
Difficulties in performing tasks that require reading						
Not at all (ref.)	1.00	1.00	1.00	1.00	1.00	1.00
Fairly little	**1.37 (1.18–1.60)**	**1.19 (1.13–1.26)**	**1.45 (1.35–1.56)**	**1.39 (1.19–1.64)**	**1.19 (1.13–1.26)**	**1.40 (1.29–1.51)**
Quite a lot or very much	**1.96 (1.65–2.33)**	**1.37 (1.29–1.46)**	**2.66 (2.45–2.89)**	**1.95 (1.62–2.35)**	**1.35 (1.26–1.45)**	**2.53 (2.31–2.76)**
Difficulties in performing tasks that require calculation						
Not at all (ref.)	1.00	1.00	1.00	1.00	1.00	1.00
Fairly little	1.13 (0.96–1.34)	**1.19 (1.13–1.26)**	**1.44 (1.33–1.56)**	1.12 (0.94–1.33)	**1.18 (1.11–1.25)**	**1.39 (1.27–1.51)**
Quite a lot or very much	**1.68 (1.41–1.99)**	**1.33 (1.25–1.41)**	**2.32 (2.14–2.53)**	**1.66 (1.39–1.99)**	**1.33 (1.24–1.42)**	**2.23 (2.04–2.44)**
(2) Difficulties in oral presentations						
Not at all (ref.)	1.00	1.00	1.00	1.00	1.00	1.00
Fairly little	0.90 (0.75–1.08)	0.97 (0.91–1.03)	**0.83 (0.77–0-91)**	0.93 (0.77–1.12)	0.95 (0.89–1.02)	**0.82 (0.75–0.90)**
Quite a lot or very much	1.12 (0.94–1.34)	0.97 (0.91–1.03)	**1.13 (1.04–1.23)**	1.14 (0.95–1.38)	0.95 (0.89–1.01)	1.09 (0.99–1.19)
(3) Difficulties in using study devices						
Not at all (ref.)	1.00	1.00	1.00	1.00	1.00	1.00
Fairly little	1.16 (1.01–1.35)	**1.08 (1.03–1.13)**	1.02 (0.96–1.10)	1.15 (0.98–1.33)	**1.09 (1.03–1.15)**	1.03 (0.96–1.11)
Quite a lot or very much	**1.92 (1.58–2.34)**	1.09 (1.01–1.18)	**1.39 (1.26–1.54)**	**1.81 (1.46–2.23)**	1.09 (1.00–1.18)	**1.37 (1.23–1.53)**

OR = odds ratio, 95% CI = 95% confidence interval. Limit of statistical significance <0.01. Results in bold are statistically significant. Reference category visits: no, there was no need for it. Multivariate model: parents’ education and unemployment during the last 12 months and country of birth (Finnish or foreign-born).

**Table 7 children-08-00647-t007:** Univariate and multivariate odds ratios (OR) for boys, based on successful access too school nurse health services, schooling difficulties and possible confounding factors, as recorded in the 2017 School Health Promotion study 2017.

Visited the School Health Nurse, Other Than for a Regular Check-Up						
	Univariate Model			Multivariate Model		
	No, I Tried but Could Not Get an Appointment OR (95% CI)	Yes, 1–2 Times OR (95% CI)	Yes, More Than 2 Times OR (95% CI)	No, I Tried but Could Not Get an Appointment OR (95% CI)	Yes, 1–2 Times OR (95% CI)	Yes, More Than 2 Times OR (95% CI)
(1) Self-reported learning difficulties						
Difficulties in following the teaching in class						
Not at all (ref.)	1.00	1.00	1.00	1.00	1.00	1.00
Fairly little	**1.48 (1.23–1.77)**	**1.33 (1.25–1.40)**	**1.49 (1.33–1.67)**	**1.46 (1.20–1.78)**	**1.31 (1.23–1.39)**	**1.55 (1.37–1.76)**
Quite a lot or very much	**2.70 (2.19–3.34)**	**1.50 (1.38–1.62)**	**3.24 (2.85–3.69)**	**2.33 (1.84–2.95)**	**1.48 (1.36–1.61)**	**2.84 (2.45–3.28)**
Difficulties in doing homework or other school tasks						
Not at all (ref.)	1.00	1.00	1.00	1.00	1.00	1.00
Fairly little	1.20 (1.00–1.45)	**1.29 (1.22–1.37)**	**1.52 (1.35–1.72)**	1.21 (0.99–1.48)	**1.27 (1.20–1.36)**	**1.58 (1.38–1.80)**
Quite a lot or very much	**2.13 (1.75–2.59)**	**1.42 (1.33–1.52)**	**3.12 (2.75–3.54)**	**1.99 (1.60–2.48)**	**1.41 (1.31–1.52)**	**2.88 (2.51–3.31)**
Difficulties in preparing for exams						
Not at all (ref.)	1.00	1.00	1.00	1.00	1.00	1.00
Fairly little	1.25 (1.02–1.53)	**1.30 (1.21–1.38)**	**1.27 (1.11–1.45)**	1.22 (0.98–1.52)	**1.27 (1.19–1.36)**	**1.30 (1.12–1.50)**
Quite a lot or very much	**1.97 (1.60–2.42)**	**1.53 (1.42–1.64)**	**2.56 (2.25–2.92)**	**1.89 (1.51–2.37)**	**1.52 (1.41–1.64)**	**2.44 (2.12–2.82)**
Difficulties in performing tasks that require writing						
Not at all (ref.)	1.00	1.00	1.00	1.00	1.00	1.00
Fairly little	1.12 (0.93–1.35)	**1.15 (1.08–1.21)**	**1.20 (1.06–1.34)**	1.05 (0.86–1.28)	**1.13 (1.06–1.20)**	**1.21 (1.07–1.37)**
Quite a lot or very much	**1.96 (1.60–2.39)**	**1.26 (1.17–1.35)**	**2.30 (2.04–2.61)**	**1.86 (1.50–2.32)**	**1.25 (1.16–1.35)**	**2.10 (1.83–2.41)**
Difficulties in performing tasks that require reading						
Not at all (ref.)	1.00	1.00	1.00	1.00	1.00	1.00
Fairly little	0.98 (0.82–1.17)	**1.15 (1.09–1.22)**	1.16 (1.03–1.29)	0.97 (0.80–1.18)	**1.14 (1.07–1.21)**	1.14 (1.01–1.29)
Quite a lot or very much	**1.88 (1.55–2.27)**	**1.34 (1.25–1.44)**	**2.34 (2.08–2.64)**	**1.76 (1.42–2.18)**	**1.34 (1.24–1.44)**	**2.06 (1.80–2.36)**
Difficulties in performing tasks that require calculation						
Not at all (ref.)	1.00	1.00	1.00	1.00	1.00	1.00
Fairly little	1.18 (0.99–1.42)	**1.17 (1.11–1.24)**	**1.32 (1.18–1.47)**	1.13 (0.92–1.38)	**1.17 (1.10–1.24)**	**1.39 (1.23–1.57)**
Quite a lot or very much	**2.20 (1.81–2.68)**	**1.25 (1.17–1.35)**	**2.27 (2.01–2.57)**	**2.10 (1.69–2.60)**	**1.23 (1.13–1.33)**	**2.05 (1.79–2.35)**
(2) Difficulties in oral presentations						
Not at all (ref.)	1.00	1.00	1.00	1.00	1.00	1.00
Fairly little	1.09 (0.90–1.32)	0.99 (0.93–1.05)	**0.84 (1.75–0.94)**	1.09 (0.89–1.34)	0.97 (0.91–1.04)	**0.84 (0.74–0.95)**
Quite a lot or very much	**1.40 (1.14–1.71)**	1.05 (0.98–1.12)	**1.24 (1.10–1.39)**	1.31 (1.05–1.64)	1.01 (0.94–1.09)	1.09 (0.96–1.25)
(3) Difficulties in using study devices						
Not at all (ref.)	1.00	1.00	1.00	1.00	1.00	1.00
Fairly little	**1.33 (1.12–1.57)**	0.99 (0.93–1.04)	1.01 (0.91–1.12)	1.19 (0.99–1.44)	0.98 (0.92–1.04)	0.99 (0.88–1.11)
Quite a lot or very much	**2.62 (2.11–3.25)**	0.92 (0.84–1.02)	**1.97 (1.71–2.27)**	**1.90 (1.48–2.45)**	0.90 (0.81–1.00)	**1.54 (1.30–1.81)**

OR = odds ratio, 95% CI = 95% confidence interval. Limit of statistical significance <0.01. Results in bold are statistically significant. Reference category visits: no, there was no need for it. Multivariate model: parents’ education and unemployment during the last 12 months and country of birth (Finnish or foreign-born).

## Data Availability

Not applicable.

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
