# Peer review of "Self-Reported School Difficulties and the Use of the School Nurse Services by Adolescent Students"

_children, 2021, doi:10.3390/children8080647_

Round 1

Reviewer 1 Report

The main objective of the paper submitted for review is to analyze the relationship between self-reported school difficulties and the use of school health nursing services by eighth and ninth grade students.

The work can make an important contribution by making some minor modifications.

Firstly, in the introduction it would be interesting to emphasize what is new about the study in comparison with other studies already carried out on the subject. 

Why do the authors believe that, with school health nurses being free of charge, lower socio-economic students have less access?

Participants: the description of the participants would be better added in the materials and method section (2.1 Data and participants) rather than in the results section. 

On the other hand, was consideration given to what the school nurse was visited for? That is, for those pupils who visited the school nurse, for what purpose did they do so?

What do the authors believe may account for the identification of a higher number of self-reported difficulties in their study compared to previous studies? 

Author Response

Thank you for the important comments. Our responses are in red.

Firstly, in the introduction it would be interesting to emphasize what is new about the study in comparison with other studies already carried out on the subject.

We write in the introduction:  "Few studies have reported the associations between adolescents’ self-reported difficulties at school and their use of school health services." We added: "This study brings new knowledge on the subject and also explores the differences between boys and girls."

Why do the authors believe that, with school health nurses being free of charge, lower socio-economic students have less access?

Kivimäki's study has found that adolescents with an immigrant background had a higher risk for perceiving the access as difficult in Finland [30]. We changed the sentence on line 101 as follows: “In Finland, students with an immigrant background, lower school well-being, lack of study support and greater health needs have reported more difficulties in access to school health nurses [30].”

Participants: the description of the participants would be better added in the materials and method section (2.1 Data and participants) rather than in the results section.

We decided to leave the description of the participants in the Results section, because we are analyzing the association between self-reported schooling difficulties and the use of nurse services when the data were controlled for background factors.

On the other hand, was consideration given to what the school nurse was visited for? That is, for those pupils who visited the school nurse, for what purpose did they do so?

This is good consideration. We added to the discussion: “However, we do not know for what purpose do the respondents visited the school health nurse.”

What do the authors believe may account for the identification of a higher number of self-reported difficulties in their study compared to previous studies?

We have considered this in this article as follows: “This may have been due to the definition, delineation, and measurement of the concept of learning disabilities. In the present study, the students’ self-reported any difficulties in schooling and learning, but their assessment of what those terms meant did not necessarily meet the diagnostic criteria of the classifications in the International Classification of Diseases, Tenth Revision.”

This manuscript has been proofread by a native English speaking professional translator with scientific proofreading competence.

Thank you very much for your good comments.

Reviewer 2 Report

Dear authors! Congratulations for you amazing study. I just have a few suggestions: 

Introduction: 

  • When you have more than 3 references sequencialy presented, you can write [5-9], instead of [5,6,7,8,9] as in line 41.
  • Maybe it would be nice for you to explain how school nursing services are provided and organized in Finland. The questions to the participants only ask if they quoting: "Visited the school health nurse, other than for a regular checkup". Don't school nurse visit the students for health education activities to health promotion?

Discussion: 

The results show that more than half of the students don't visit the school nurse, other than for a regular checkup. Why? Because just the ones who have problems visit the nurse as you refer? The others don't see the nurse as a reference to other issues beside problem resolution? 

Congratulations for your important study! :) 

Author Response

Thank you very much for your encouraging words and good comments! Our responses are in red.

Introduction: 

  • When you have more than 3 references sequencialy presented, you can write [5-9], instead of [5,6,7,8,9] as in line 41.

Good attention, done.

  • Maybe it would be nice for you to explain how school nursing services are provided and organized in Finland. The questions to the participants only ask if they quoting: "Visited the school health nurse, other than for a regular checkup". Don't school nurse visit the students for health education activities to health promotion?

We decide to add:

“In Finland school health services are part of universal preventive health care. Services are organized municipally and include health examinations, health care counselling, acute medical care and promotion of well-being and safety at school.”

References:

Health Care Act 1326/2010. Ministry of Social Affairs and Health, Finland. Finlex, http://www.finlex.fi/en/laki/kaannokset/2010/en20101326.pdf (last accessed 21 July 2021). 

Government Decree 338/2011 on maternity and child health clinic services, school and student health services and preventive oral health services for children and youth. Finlex, http://www.finlex.fi/en/laki/kaannokset/2011/en20110338.pdf, (last accessed 21 July 2021). 

Discussion: 

The results show that more than half of the students don't visit the school nurse, other than for a regular checkup. Why? Because just the ones who have problems visit the nurse as you refer? The others don't see the nurse as a reference to other issues beside problem resolution? 

We added to the discussion: “However, we do not know for what purpose do the respondents visited the school health nurse.”

This manuscript has been proofread by a native English speaking professional translator with scientific proofreading competence.

Thank you very much!